# White-box Error Correction Code Transformer

**Ziyan Zheng**[1,2,4]    **Chin Wa (Ken) Lau**[2,3]    **Nian Guo**[2]    **Xiang Shi**[2,4]    **Shao-Lun Huang**[1]*

[1]Tsinghua Shenzhen International Graduate School    [2]Huawei Technologies Co., Ltd.
[3]The Chinese University of Hong Kong    [4]Tsinghua University
zhengzy7@mails.tsinghua.edu.cn    kenlau@ie.cuhk.edu.hk    guonian4@huawei.com
shixiang23@huawei.com    shaolun.huang@sz.tsinghua.edu.cn

Error correcting codes (ECCs) play a crucial role in modern communication systems by ensuring reliable data transmission over noisy channels. While traditional algorithms based on belief propagation suffer from limited decoding performance, transformer-based approaches have emerged as powerful solutions for ECC decoding. However, the internal mechanisms of transformer-based approaches remain largely unexplained, making it challenging to understand and improve their performance. In this paper, we propose a White-box Error Correction Code Transformer (WECCT) that provides theoretical insights into transformer-based decoding. By formulating the decoding problem from a sparse rate reduction perspective and introducing a novel Multi-head Tanner-subspaces Self Attention mechanism, our approach provides a parameter-efficient and theoretically principled framework for understanding transformer-based decoding. Extensive experiments across various code families demonstrate that this interpretable design achieves competitive performance compared to state-of-the-art decoders.

## 1. Introduction

Error correcting codes (ECCs) are fundamental building blocks in modern communication systems, enabling reliable data transmission across noisy channels by adding redundant information to the transmitted messages [1, 2]. The theoretical foundation of error correction coding was established by Shannon's seminal work [3], which proved the existence of codes capable of achieving reliable communication up to channel capacity. However, the constructive design of practical codes and efficient decoders remains a significant challenge [4]. While optimal decoding is theoretically defined by the maximum likelihood (ML) rule, its implementation is NP-hard for general linear codes, necessitating the development of efficient approximate solutions [5]. This challenge has become increasingly critical with the growing demands of modern applications, from high-speed wireless communications to deep space exploration, where both reliability and computational efficiency are paramount [6, 7]. The complexity of ML decoding arises from the combinatorial nature of the problem: for a code of length $n$, the decoder must effectively search through a space of $2^k$ possible codewords, where $k < n$ is the information length. This exponential complexity makes ML decoding impractical for most real-world applications, particularly for respectively longer codes where error correction is most needed [8].

**Classical Approaches.** The development of decoding algorithms has seen several major paradigm shifts over the past decades. A significant breakthrough came with belief propagation (BP) and message-passing algorithms, particularly for low-density parity-check (LDPC) codes [1]. BP operates by iteratively exchanging probabilistic messages between variable nodes and check nodes in the code's Tanner graph, providing a practical approach to approximate ML decoding [9]. While widely adopted in modern communication standards [6], BP suffers from limitations such as uncertain convergence and performance degradation with short cycles in the Tanner graph [1]. For the most recent decade, the development of deep learning has introduced two main approaches to neural decoding. Model-based neural decoders enhance BP by parameterizing message-passing operations with neural networks [10–12], maintaining interpretability while learning optimal update rules. However, the fixed message-passing scheme and local nature of updates may prevent these decoders from discovering more efficient global decoding strategies and achieving satisfac-

---

*Corresponding author

Second Conference on Parsimony and Learning (CPAL 2025).

tory results [11]. Model-free neural decoders, in contrast, employ generic neural architectures or fully-connected networks without explicit reliance on traditional decoding algorithms [13, 14].

**Transformer-based Methods.** The landscape of neural decoding was fundamentally transformed with the introduction of transformer-based architectures. The Error Correction Code Transformer (ECCT) [15] pioneered this direction by adapting the transformer architecture [16] for ECC decoding. At its core, ECCT processes concatenated magnitude and syndrome vectors through masked self-attention modules, where the interaction between tokens follows the code's parity check matrix. While achieving competitive performance, ECCT's design faces limitations that the concatenated representation may not optimally leverage the distinct properties of magnitude and syndrome information. Building on ECCT's success, the Cross-attention Message-Passing Transformer (CrossMPT) [17] processes magnitude and syndrome vectors separately through cross-attention blocks, better reflecting their distinct roles in error correction. By sharing operational principles with traditional message-passing decoders, CrossMPT achieves improved performance through specialized information processing. However, CrossMPT still relies on heuristic masking schemes, and its theoretical foundation remains unexplored - particularly regarding the nature of information learned in the latent space and passed among nodes. This lack of interpretability presents a significant barrier to understanding the model's behavior.

**White-box Transformer.** More recently, Yu et al. [18] proposed the Coding-RATE transformer (CRATE), a White-box Transformer architecture that provides theoretical insights into transformer models through the lens of data compression. They show that transformer architectures can be interpreted as optimizing a sparse rate reduction objective: the modified multi-head self-attention implements approximate gradient descent on the coding rate to compress representations, while the following feed-forward networks promote sparsity in the learned features. This framework provides clear theoretical justification for each architectural component.

**Our Contributions.** In this work, we propose a White-box Error Correction Code Transformer (WECCT) framework that builds upon CRATE's theoretical foundation while specifically targeting the challenges of ECC decoding. Our WECCT represents the first attempt to introduce an interpretable, white-box transformer architecture for decoding tasks. We firstly design a novel Multi-head Tanner-subspaces Self Attention (MTSA) mechanism that integrates code structure into representation learning, enabling structured message passing between bits and syndromes through Tanner subspaces. We further propose a two-stage optimization framework where attention operations implement rate reduction through MTSA and feed-forward layers promote structured sparsity through Iterative Shrinkage-Thresholding Algorithm (ISTA). This theoretically-motivated design provides clear mathematical objectives and bridges the gap between transformer architectures and coding theory. The resulting decoder not only substantially promotes parametric efficiency, but also achieves competitive performance across various code families while maintaining its interpretability.

**Paper Organization.** The remainder of this paper is organized as follows. Section 2 provides necessary background on ECCs and CRATE. Section 3 presents our theoretical framework connecting transformer-based decoding with sparse rate reduction principles and describes the proposed WECCT architecture and algorithm in detail. Section 4 presents experimental results and analysis and Section 5 finally concludes the paper.

**Notation.** Throughout this paper, scalars are written as non-bold letters (e.g., $x, n, k$), vectors as lower-case bold letters (e.g., $\boldsymbol{x}, \boldsymbol{y}, \boldsymbol{z}$), and matrices as upper-case letters (e.g., $G, H, \boldsymbol{U}, \boldsymbol{D}$). We use $\mathbf{1}$ to represent a vector or matrix of all ones with appropriate dimensions, and $\boldsymbol{I}$ to represent an identity matrix. Calligraphic letters (e.g., $\mathcal{T}$) indicate spaces or subspaces. For a matrix $\boldsymbol{A}$, $\boldsymbol{A}^*$ represents its transpose, and $\boldsymbol{A}_{ij}$ its $(i, j)$-th entry. For a real vector $\boldsymbol{x}$, $\boldsymbol{x}_i$ is its $i$-th element, $\|\boldsymbol{x}\|_p$ is its $\ell_p$ norm, and $\text{sign}(\boldsymbol{x})$ applies the sign function element-wise. For a binary vector $\boldsymbol{x}$, $\text{bin}(\boldsymbol{x})$ converts its elements from $\{\pm 1\}$ to $\{0, 1\}$. The notation $[n]$ refers to the set $\{1, \ldots, n\}$. For a set $S$, the notation $S^n$ denotes a n-dimensional column vector with elements in $S$. Note that $k$ denotes the information length of the code when used in coding context, while it represents the index of attention heads or subspace bases in transformer context following standard notation - these should not be confused despite using the same symbol. All logarithms are natural logarithms unless otherwise specified.

## 2. Background

### 2.1. Error Correction Codes

Let $\mathcal{C}$ be a linear block code defined by a generator matrix $G \in \{0,1\}^{n \times k}$ and a parity check matrix $H \in \{0,1\}^{(n-k) \times n}$, where $k$ and $n$ are the information length and code length respectively [2]. These matrices satisfy $HG = 0$ over the binary field $\mathbb{F}_2$. A message $\boldsymbol{m} \in \{0,1\}^k$ is encoded into a codeword $\boldsymbol{x} \in \mathcal{C} \subset \{0,1\}^n$ through $\boldsymbol{x} = G\boldsymbol{m}$. The codeword is modulated using Binary Phase Shift Keying (BPSK) to obtain $\boldsymbol{x}_s \in \{\pm 1\}^n$ before transmission over the channel. In this work, we consider the Additive White Gaussian Noise (AWGN) channel, where the received signal $\boldsymbol{y} \in \mathbb{R}^n$ is given by $\boldsymbol{y} = \boldsymbol{x}_s + \boldsymbol{z}$ where $\boldsymbol{z} \sim \mathcal{N}(0, \sigma^2 \mathbf{1}_n)$ represents the channel noise with variance $\sigma^2$.

The optimal Maximum Likelihood (ML) decoder aims to find the most likely transmitted codeword given the received signal:

$$\hat{\boldsymbol{x}} = \mathrm{argmax}_{\boldsymbol{x} \in C} \ p(\boldsymbol{y}|\boldsymbol{x}). \tag{1}$$

The general decoders take as input the received signal $\boldsymbol{y}$ and syndrome vector $s(\boldsymbol{y}) = H\boldsymbol{y}_b$, where $\boldsymbol{y}_b = \mathrm{bin}(\mathrm{sign}(\boldsymbol{y}))$ represents the hard decision on $\boldsymbol{y}$, with $\mathrm{bin}(+1) = 0$ and $\mathrm{bin}(-1) = 1$. Note that the received signal contains both sign and reliability information for individual bits, while the syndrome vector indicates relationships between potentially erroneous positions [1, 17].

The relationships among bits and syndromes can be visualized using a Tanner graph [19], which is a bipartite graph representation of the parity check matrix $H$. The graph consists of $n$ variable nodes (representing codeword bits) and $n - k$ check nodes (representing parity check equations). An edge exists between variable node $i$ and check node $j$ if and only if $H_{ji} = 1$. Traditional BP operates by passing messages along these edges, where each message represents the probability or log-likelihood ratio of a bit being 0 or 1 [1]. The algorithm iteratively updates the messages until convergence or a maximum number of iterations is reached. This Tanner graph structure also plays a crucial role in our proposed attention mechanism, as we will see in Section 3.1.

### 2.2. White-box Transformer via Sparse Rate Reduction

The theoretical understanding of transformers has been significantly advanced by CRATE [18], which showed that the key components of transformer architectures can be derived from the principle of rate reduction. Given a set of tokens $\boldsymbol{Z} = [\boldsymbol{z}_1, ..., \boldsymbol{z}_n] \in \mathbb{R}^{d \times n}$ ($n$ tokens with dimension $d$ of each token), it formulates the learning objective as maximizing the sparse rate reduction: $\max_f \mathbb{E}_{\boldsymbol{Z}=f(\boldsymbol{X})} \left[ R(\boldsymbol{Z}) - R^c(\boldsymbol{Z} \mid \boldsymbol{U}_{[K]}) - \lambda \|\boldsymbol{Z}\|_0 \right]$, where $R(\boldsymbol{Z}) = \frac{1}{2} \log \det(\boldsymbol{I} + \alpha \boldsymbol{Z}^* \boldsymbol{Z})$ is the coding rate of the whole token set measuring the overall information content, $R^c(\boldsymbol{Z} \mid \boldsymbol{U}_{[K]}) = \frac{1}{2} \sum_{k=1}^{K} \log \det(\boldsymbol{I} + \beta(\boldsymbol{U}_k^* \boldsymbol{Z})^*(\boldsymbol{U}_k^* \boldsymbol{Z}))$ is the coding rate when tokens are encoded by a mixture of $K$ low-dimensional subspaces with bases $\boldsymbol{U}_{[K]} = (\boldsymbol{U}_k)_{k=1}^K$ in which $\boldsymbol{U}_k \in \mathbb{R}^{d \times p}$; $\alpha = d/n\epsilon^2, \beta = p/n\epsilon^2$ with quantization precision $\epsilon > 0$, and $\lambda \|\boldsymbol{Z}\|_0 \geq 0$ promotes sparsity. This framework implements compression through Multi-head Subspaces Self Attention and sparsification through ISTA, providing a complete theoretical interpretation of transformer layers. While CRATE provides valuable insights into why transformers are effective at learning compact representations, our work focuses on adapting these principles specifically for ECCs, where the goal is to promote reliable decoding via learning robust and interpretable representations for the bits and syndromes.

## 3. WECCT Framework

In this section, we present our WECCT architecture that is theoretically derived from the principle of sparse rate reduction. Section 3.1 introduces our novel MTSA mechanism for feature compression, Section 3.2 describes the sparsification process via ISTA, and Section 3.3 details the complete architecture design.

For the AWGN channel, where the noise follows a Gaussian distribution, the decoding objective (1) is equivalent to minimizing the Euclidean distance [1]:

$$\hat{\boldsymbol{x}}_s = \underset{\boldsymbol{x} \in \mathcal{C}}{\mathrm{argmin}} \ \|\boldsymbol{y} - \boldsymbol{x}_s\|_2^2. \tag{2}$$

The optimal local solution can be characterized by $\mathbb{E}[\boldsymbol{x}_s|\boldsymbol{y}]$. Through Tweedie's formula [20], this conditional expectation can be expressed as a denoising process:

$$\mathbb{E}[\boldsymbol{x}_s|\boldsymbol{y}] = \boldsymbol{y} + \sigma^2 \nabla \log p(\boldsymbol{y}), \tag{3}$$

where $\nabla \log p(\boldsymbol{y})$ is the score function. For linear block codes with parity check matrix $H$, the code structure is characterized by the constraint $H\boldsymbol{x} = \boldsymbol{0}$ for any valid codeword $\boldsymbol{x}$. Therefore, optimal decoding requires joint denoising that respects both the local noise statistics and the global structural relationship. We map the received signal $\boldsymbol{y}$ to bit representations $\boldsymbol{Z}_b \in \mathbb{R}^{d \times n}$ and its syndrome $s(\boldsymbol{y})$ to syndrome representations $\boldsymbol{Z}_s \in \mathbb{R}^{d \times (n-k)}$, where each column corresponds to a token embedding in a $d$-dimensional space. Let $\boldsymbol{Z} = [\boldsymbol{Z}_b, \boldsymbol{Z}_s] \in \mathbb{R}^{d \times (2n-k)}$ combine these representations into a unified space. Through energy-based insights [21], We formalize this intuition as sparse rate reduction objective on token representations:

**Approximation 1** (From ML Decoding to Sparse Rate Reduction)**.** *The ML decoding objective can be approximated by optimizing a sparse rate reduction objective over the joint representation space of bits and syndromes:*

$$\max_f \mathbb{E}_{\boldsymbol{Z}} \left[ R(\boldsymbol{Z}) - R^c(\boldsymbol{Z} \mid \boldsymbol{U}_{[K]}) - \lambda \|\boldsymbol{Z}\|_1 \right], \tag{4}$$

*where the coding rate measure $R(\cdot)$, $R^c(\cdot \mid \cdot)$ and the subspace bases $\boldsymbol{U}_{[K]}$ are defined in Section 2.2.*

*Proof:* See Appendix A.

Through batch training, we can approximate the expectations by empirical averages, allowing us to omit the expectation notation in the following formulations. The optimization is split across two key components of our architecture. Section 3.1 focuses on optimizing the feature compression (5) term through MTSA:

$$\min_f R^c(\boldsymbol{Z} \mid \boldsymbol{U}_{[K]}), \tag{5}$$

while Section 3.2 addresses the sparsification objective (6) through ISTA:

$$\min_f \lambda \|\boldsymbol{Z}\|_1 - R(\boldsymbol{Z}). \tag{6}$$

This decomposition aligns with the theoretical framework in [18], where transformer layers alternate between feature compression and sparsification steps.

## 3.1. Multi-head Tanner-subspaces Self Attention

To optimize the objective (5) while respecting the structural constraints of error correction codes, we introduce a novel MTSA mechanism. The key insight of MTSA is to incorporate the Tanner graph structure of the code into the attention computation, ensuring that information only flows between connected bit and syndrome nodes. We first formally define the notion of Tanner subspaces that captures this connectivity structure:

**Definition 1** (Tanner Subspaces)**.** *Let $H \in \{0,1\}^{(n-k) \times n}$ be a parity check matrix. Let $\boldsymbol{Z} = [\boldsymbol{Z}_b, \boldsymbol{Z}_s] = [\boldsymbol{z}_1, ..., \boldsymbol{z}_{2n-k}] \in \mathbb{R}^{d \times (2n-k)}$ be the d-dimensional representations of both bits and syndromes. For each node $i \in [2n-k]$, the Tanner subspace $\mathcal{T}_i$ is defined as:*

$$\mathcal{T}_i \triangleq \mathrm{Span}\{\boldsymbol{z}_j \mid \mathcal{M}(H)_{ji} = 1\} \subset \mathbb{R}^d, \tag{7}$$

*where $\mathcal{M}(H)$ is the extended connectivity matrix:*

$$\mathcal{M}(H) \triangleq \begin{bmatrix} \boldsymbol{0}_{n \times n} & H^* \\ H & \boldsymbol{0}_{(n-k) \times (n-k)} \end{bmatrix}. \tag{8}$$

This definition establishes a geometric interpretation of the Tanner graph structure through subspaces. Each representation $\boldsymbol{z}_i$ resides in a subspace $\mathcal{T}_i$ that is spanned by its connected nodes in the Tanner graph. These Tanner subspaces naturally encode the local connectivity of the code's Tanner graph in a geometric manner: two nodes can directly interact in the representation space if and only

if they share a common subspace, which occurs precisely when they are connected in the Tanner graph (i.e., $\mathcal{M}(H)_{ji} = 1$). This geometric interpretation provides a principled way to constrain information flow in our model and ensures that the learned representations inherently respect the algebraic structure of the code.

Building on these Tanner subspaces, we develop our MTSA mechanism that efficiently implements gradient descent on the coding rate while preserving the Tanner graph structure:

**Approximation 2** (Multi-head Tanner-subspaces Self Attention). *Let $\boldsymbol{Z} \in \mathbb{R}^{d \times (2n-k)}$ have unit-norm columns, and $\boldsymbol{U}_{[K]} = (\boldsymbol{U}_1, \ldots, \boldsymbol{U}_K)$ such that each $\boldsymbol{U}_k \in \mathbb{R}^{d \times p}$ is an orthogonal matrix, the $(\boldsymbol{U}_k)_{k=1}^K$ are incoherent, and the Tanner subspaces $\mathcal{T} = \{\mathcal{T}_i, i \in [2n-k]\}$ are induced by $\mathcal{M}(H)$. The columns $\boldsymbol{z}_i$ approximately lie on $(\bigcup_{k=1}^K \operatorname{Span}(\boldsymbol{U}_k)) \cap \mathcal{T}_i$. Let $\kappa > 0$. Then the gradient descent could be calculated as:*

$$\boldsymbol{Z} - \kappa \nabla_{\boldsymbol{Z}} R^c(\boldsymbol{Z} \mid \boldsymbol{U}_{[K]}) \approx (1 - \kappa\beta)\boldsymbol{Z} + \kappa\beta \operatorname{MTSA}(\boldsymbol{Z} \mid \boldsymbol{U}_{[K]}), \tag{9}$$

*where*

$$
\begin{aligned}
\operatorname{TSA}(\boldsymbol{Z} \mid \boldsymbol{U}_k) &\triangleq (\boldsymbol{U}_k^* \boldsymbol{Z}) \operatorname{softmax}((\boldsymbol{U}_k^* \boldsymbol{Z})^*(\boldsymbol{U}_k^* \boldsymbol{Z}) + \phi(\mathcal{M}(H))), \\
\operatorname{MTSA}(\boldsymbol{Z} \mid \boldsymbol{U}_{[K]}) &\triangleq \beta[\boldsymbol{U}_1, \ldots, \boldsymbol{U}_K] \begin{bmatrix} \operatorname{TSA}(\boldsymbol{Z} \mid \boldsymbol{U}_1) \\ \vdots \\ \operatorname{TSA}(\boldsymbol{Z} \mid \boldsymbol{U}_K) \end{bmatrix},
\end{aligned}
\tag{10}
$$

*where* $\operatorname{softmax}(\cdot)$ *is the softmax operator* (*applied to each column of an input matrix*), *i.e.,*

$$\operatorname{softmax}(\boldsymbol{v}) = \frac{1}{\sum_{i=1}^n e^{v_i}} \begin{bmatrix} e^{v_1} \\ \vdots \\ e^{v_n} \end{bmatrix}, \tag{11}$$

$$\operatorname{softmax}([\boldsymbol{v}_1, \ldots, \boldsymbol{v}_K]) = [\operatorname{softmax}(\boldsymbol{v}_1), \ldots, \operatorname{softmax}(\boldsymbol{v}_K)], \tag{12}$$

*and the masking function* $\phi(\mathcal{M}(H))$ *is defined as*

$$\phi(\mathcal{M}(H)) \triangleq \begin{bmatrix} -\boldsymbol{\infty}_{n \times n} & \phi(H^*) \\ \phi(H) & -\boldsymbol{\infty}_{(n-k) \times (n-k)} \end{bmatrix}, \tag{13}$$

*where* $\phi : \{0,1\}^{m \times n} \to \{-\infty, 0\}^{m \times n}$ *is an element-wise operator that maps 0 entries to* $-\infty$ *and 1 entries to 0, ensuring attention only flows among connected bit and syndrome nodes in the Tanner graph.*

*Proof:* See Appendix B. We first derive the exact gradient and then approximate it using von Neumann expansion, while consideration on Tanner subspaces finally leads to our MTSA formulation.

The operations are then simplified to the form $\boldsymbol{Z}^{l+1/2} = \boldsymbol{Z}^l + \operatorname{MTSA}(\boldsymbol{Z}^l \mid \boldsymbol{U}_{[K]}^l)$, where $l$ denotes the layer index and $l + 1/2$ denotes the intermediate output layer after MTSA.

## 3.2. Sparse Coding via ISTA

After the Tanner-subspace attention update, we optimize the sparsification term (6) following [18]:

$$\boldsymbol{Z}^{l+1} \approx \underset{\boldsymbol{Z}}{\operatorname{argmin}} \left\{ \lambda \|\boldsymbol{Z}\|_1 + \frac{1}{2} \|\boldsymbol{Z}^{l+1/2} - \boldsymbol{D}^l \boldsymbol{Z}\|_F^2 \right\}, \tag{14}$$

where $\boldsymbol{D}^l \in \mathbb{R}^{d \times d}$ is a learnable (complete) incoherent or orthogonal dictionary that enforces $R(\boldsymbol{Z})^{l+1} \approx R(\boldsymbol{Z})^{l+1/2}$. This optimization can be solved using ISTA [22]:

$$\boldsymbol{Z}^{l+1} = \operatorname{ISTA}(\boldsymbol{Z}^{l+1/2} | \boldsymbol{D}^l) \triangleq \operatorname{ReLU}(\boldsymbol{Z}^{l+1/2} - \eta(\boldsymbol{D}^l)^*(\boldsymbol{D}^l \boldsymbol{Z}^{l+1/2} - \boldsymbol{Z}^{l+1/2}) - \eta\lambda\boldsymbol{1}), \tag{15}$$

where $\eta > 0$ is the step size. Through this structured sparse coding step, we ensure that both bit and syndrome representations maintain efficient, sparse patterns while preserving the essential information for error correction.

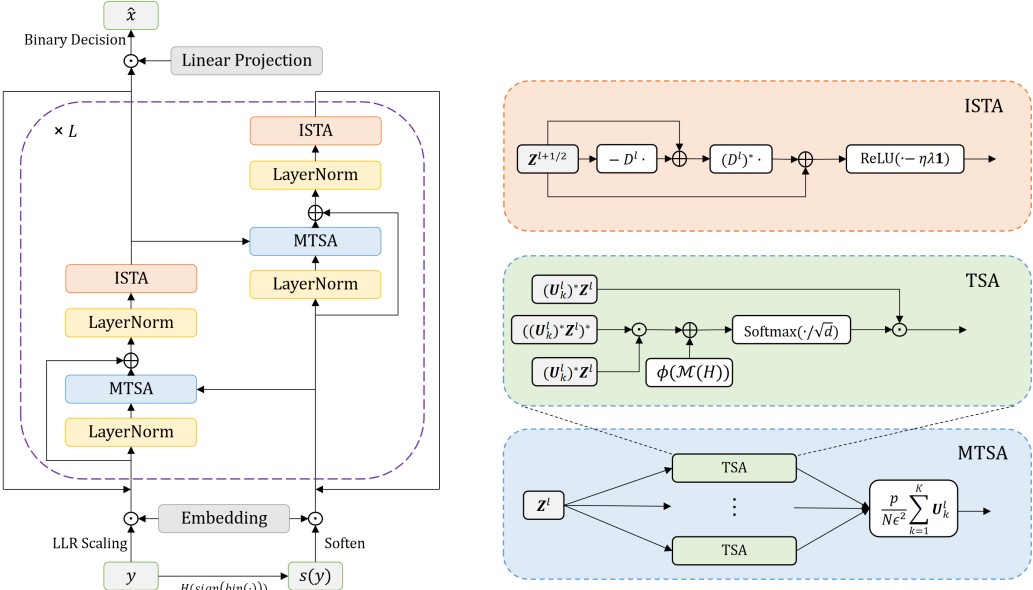

Figure 1: Overview of the WECCT architecture. The model architecture flows from bottom to top on the left, consisting of inputs embedding, decoder layers and outputs prediction, with the detailed structure of decoder layers expanded on the right.

## 3.3. The Overall Architecture

The full WECCT architecture is designed to effectively process both bit and syndrome information through a series of structured iterative transformations, enabling rich interactions through the Tanner graph structure. Figure 1 roughly illustrates the complete architecture, which consists of three main components: input embedding layer, multiple decoder layers, and output prediction layer. The detailed decoding process is discussed as follows and summarized in Algorithm 1.

For input embedding, unlike ECCT [15] and CrossMPT [17] that use signal magnitude and binary syndrome, we apply LLR scaling to the received signal itself for bit tokens and use reliability-scaled soft syndrome for syndrome tokens. Specifically, given the received signal $\boldsymbol{y} \in \mathbb{R}^n$, for bit tokens we compute:

$$\boldsymbol{y}_{\text{llr}} = 2\boldsymbol{y}/\sigma^2, \tag{16}$$

$$\boldsymbol{z}_i^0 = \boldsymbol{w}_{\text{emb},i}\boldsymbol{y}_{\text{llr},i}, \quad i = 1, \ldots, n, \tag{17}$$

where $\boldsymbol{w}_{\text{emb},i} \in \mathbb{R}^d$ is a learnable embedding vector for the $i$-th bit position, and $\sigma^2$ is the noise variance of the AWGN channel. For syndrome tokens, we compute soft syndrome by scaling each parity check equation with the magnitude of its least reliable bit:

$$\boldsymbol{s}_{\text{soft},j} = \min_{i:H_{ji}=1} |\boldsymbol{y}_i| \cdot H_j \cdot \text{bin}(\text{sign}(\boldsymbol{y}_j)), \quad j = 1, \ldots, n - k, \tag{18}$$

$$\boldsymbol{z}_{n+j}^0 = \boldsymbol{w}_{\text{emb},n+j}\boldsymbol{s}_{\text{soft},j}, \quad j = 1, \ldots, n - k, \tag{19}$$

where $\boldsymbol{w}_{\text{emb},n+j} \in \mathbb{R}^d$ is a learnable embedding vector for the $j$-th syndrome position, and $H_j$ denotes the $j$-th row of the parity check matrix.

After the initial embedding, the representations are processed through $L$ decoder layers, each implementing MTSA and ISTA operations. Within each layer $l = 0$ to $L - 1$, inspired by the alternating optimization in [17], we adopt a sequential update strategy that processes bit and syndrome domains iteratively: bit representations are first refined through MTSA and ISTA steps while holding syndrome tokens fixed, followed by syndrome updates based on the improved bit representations. This alternating denoising pattern employs domain-specific learning parameters: subspace bases $\boldsymbol{U}_{b,[K]}^l$ and $\boldsymbol{U}_{s,[K]}^l$ for bits and syndromes respectively, along with their sparsification dictionaries

---

**Algorithm 1** White-box Error Correction Code Transformer

---

**Input:** Input: $\boldsymbol{y} \in \mathbb{R}^n$, $H \in \{0,1\}^{(n-k)\times n}$, Parameters: $\{\boldsymbol{U}_{b,[K]}^l, \boldsymbol{U}_{s,[K]}^l\}_{l=0}^{L-1}$ (subspace bases), $\{\boldsymbol{D}_b^l, \boldsymbol{D}_s^l\}_{l=0}^{L-1}$ (dictionaries), $\{\boldsymbol{w}_{\text{emb},i}\}_{i=1}^{2n-k}$ (input embeddings), $\{\boldsymbol{w}_{\text{out},i}, \theta_i\}_{i=1}^n$ (output projections)

**Output:** $\hat{\boldsymbol{x}} \in \{0,1\}^n$

1: $\boldsymbol{y}_{\text{llr}} = 2\boldsymbol{y}/\sigma^2$ {LLR Scaling}
2: **for** $i = 1$ to $n$ **do**
3:    $\boldsymbol{z}_i^0 = \boldsymbol{w}_{\text{emb},i}\boldsymbol{y}_{\text{llr},i}$ {Bit Token Embedding}
4: **end for**
5: **for** $j = 1$ to $n - k$ **do**
6:    $\boldsymbol{s}_{\text{soft},j} = \min_{i:H_{ji}=1}|\boldsymbol{y}_i| \cdot H_j \cdot \text{bin}(\text{sign}(\boldsymbol{y}))$ {Soft Syndrome}
7:    $\boldsymbol{z}_{n+j}^0 = \boldsymbol{w}_{\text{emb},n+j}\boldsymbol{s}_{\text{soft},j}$ {Syndrome Token Embedding}
8: **end for**
9: $\boldsymbol{Z}_b^0 = [\boldsymbol{z}_1^0,\ldots,\boldsymbol{z}_n^0]$, $\boldsymbol{Z}_s^0 = [\boldsymbol{z}_{n+1}^0,\ldots,\boldsymbol{z}_{2n-k}^0]$
10: **for** $l = 0$ to $L - 1$ **do**
11:    $\tilde{\boldsymbol{Z}}_b^l = \text{LayerNorm}(\boldsymbol{Z}_b^l)$
12:    $\boldsymbol{Z}_b^{l+1/2} = \boldsymbol{Z}_b^l + \text{MTSA}([\tilde{\boldsymbol{Z}}_b^l, \boldsymbol{Z}_s^l] \mid \boldsymbol{U}_{b,[K]}^l)_{[:,1:n]}$ with $H$
13:    $\hat{\boldsymbol{Z}}_b^{l+1/2} = \text{LayerNorm}(\boldsymbol{Z}_b^{l+1/2})$
14:    $\boldsymbol{Z}_b^{l+1} = \boldsymbol{Z}_b^{l+1/2} + \text{ISTA}(\hat{\boldsymbol{Z}}_b^{l+1/2}|\boldsymbol{D}_b^l)$
15:    $\tilde{\boldsymbol{Z}}_s^l = \text{LayerNorm}(\boldsymbol{Z}_s^l)$
16:    $\boldsymbol{Z}_s^{l+1/2} = \boldsymbol{Z}_s^l + \text{MTSA}([\boldsymbol{Z}_b^{l+1}, \tilde{\boldsymbol{Z}}_s^l] \mid \boldsymbol{U}_{s,[K]}^l)_{[:,n+1:2n-k]}$ with $H$
17:    $\hat{\boldsymbol{Z}}_s^{l+1/2} = \text{LayerNorm}(\boldsymbol{Z}_s^{l+1/2})$
18:    $\boldsymbol{Z}_s^{l+1} = \boldsymbol{Z}_s^{l+1/2} + \text{ISTA}(\hat{\boldsymbol{Z}}_s^{l+1/2}|\boldsymbol{D}_s^l)$
19: **end for**
20: **for** $i = 1$ to $n$ **do**
21:    $p_i = \boldsymbol{w}_{\text{out},i}^* \boldsymbol{z}_i^L + \boldsymbol{\theta}_i$ {Linear Projection}
22: **end for**
23: $\hat{\boldsymbol{x}} = \mathbb{1}[\boldsymbol{p} > 0.5]$ {Binary Decision}
24: **return** $\hat{\boldsymbol{x}}$

---

$\boldsymbol{D}_b^l$ and $\boldsymbol{D}_s^l$. This specialization allows each domain to learn distinct features that reflect their complementary roles in error correction - bits carrying the actual (local) information while syndromes providing error detection constraints.

The final bit representations $\{\boldsymbol{z}_{b,i}^L\}_{i=1}^n$ then pass through independent token-specific linear projections to generate the decoded codeword. Note that unlike previous transformer-based approaches [15, 17] that concatenate all bit and syndrome tokens to predict the multiplicative noise through a fully-connected layer, our decoder directly estimates the probability of each transmitted bit only using bit tokens. This design choice preserves the spatial structure and offers a more direct path to codeword recovery:

$$\boldsymbol{p}_i = \boldsymbol{w}_{\text{out},i}^* \boldsymbol{z}_i^L + \boldsymbol{\theta}_i, \quad i = 1,\ldots,n, \tag{20}$$

$$\hat{\boldsymbol{x}} = \mathbb{1}[\boldsymbol{p} > 0.5], \tag{21}$$

where $\boldsymbol{w}_{\text{out},i} \in \mathbb{R}^d$ is the learnable projection vector, $\boldsymbol{\theta}_i \in \mathbb{R}$ is the learnable bias term specific to the $i$-th bit position, and $\boldsymbol{p}$ is the output probability vector for binary decision.

Through this architecture, our model achieves a balance among structural awareness (via MTSA's Tanner graph constraints), representation efficiency (via ISTA's sparsity promotion), and learning capacity (via the trainable embeddings, bases, dictionaries and output projections), which enables our model to effectively capture and utilize the inherent properties of ECCs.

Table 1: Comparison of decoding performance at three different SNR values (4 dB, 5 dB, 6 dB) for different decoders, measured by the negative natural logarithm of BER (higher is better). For each specific code in the WECCT column, the first row shows results with 6 decoder layers ($L = 6$), while the second row shows results with 12 decoder layers ($L = 12$). Best results are shown in **bold** and second best results are underlined.

| Model | | BP | | | AR BP | | | ECCT | | | CrossMPT | | | WECCT | | |
|---|---|---|---|---|---|---|---|---|---|---|---|---|---|---|---|---|
| Codes | Parameter | 4 | 5 | 6 | 4 | 5 | 6 | 4 | 5 | 6 | 4 | 5 | 6 | 4 | 5 | 6 |
| BCH | (31,16) | 4.63 | 5.88 | 7.60 | 5.48 | 7.37 | 9.60 | 6.39 | 8.29 | 10.66 | **6.98** | **9.25** | 12.48 | 6.31
6.51 | 8.52
8.73 | 11.39
**12.65** |
| BCH | (63,36) | 4.03 | 5.42 | 7.26 | 4.57 | 6.39 | 8.92 | 4.86 | 6.65 | 9.10 | **5.03** | **6.91** | **9.37** | 4.81
4.91 | 6.53
6.70 | 9.01
9.24 |
| BCH | (63,45) | 4.36 | 5.55 | 7.26 | 4.97 | 6.90 | 9.41 | 5.60 | 7.79 | 10.93 | **5.90** | 8.20 | **11.62** | 5.55
5.87 | 7.80
**8.27** | 10.90
11.25 |
| BCH | (63,51) | 4.5 | 5.82 | 7.42 | 5.17 | 7.16 | 9.53 | 5.66 | 7.89 | 11.01 | **5.78** | **8.08** | **11.41** | 5.54
5.62 | 7.76
7.89 | 10.86
11.04 |
| Polar | (64,32) | 4.26 | 5.38 | 6.50 | 5.57 | 7.43 | 9.82 | 6.99 | 9.44 | 12.32 | **7.50** | **9.97** | **13.31** | 6.42
6.71 | 8.69
9.03 | 11.34
12.54 |
| Polar | (64,48) | 4.74 | 5.94 | 7.42 | 5.41 | 7.19 | 9.30 | 6.36 | 8.46 | 11.09 | **6.51** | **8.70** | 11.31 | 6.08
6.32 | 8.19
8.48 | 11.13
**11.36** |
| Polar | (128,64) | 4.1 | 5.11 | 6.15 | 4.84 | 6.78 | 9.3 | 5.92 | 8.64 | 12.18 | **7.52** | **11.21** | **14.76** | 5.43
6.11 | 7.86
8.94 | 11.20
12.32 |
| Polar | (128,86) | 4.49 | 5.65 | 6.97 | 5.39 | 7.37 | 10.13 | 6.31 | 9.01 | 12.45 | **7.51** | **10.83** | **15.24** | 6.11
6.97 | 8.83
10.22 | 12.60
14.29 |
| Polar | (128,96) | 4.61 | 5.79 | 7.08 | 5.27 | 7.44 | 10.2 | 6.31 | 9.12 | 12.47 | **7.15** | **10.15** | 13.13 | 6.09
6.48 | 8.84
9.35 | 11.96
**13.48** |
| LDPC | (49,24) | 6.23 | 8.19 | 11.72 | 6.58 | 9.39 | 12.39 | 6.13 | 8.71 | 12.10 | 6.68 | 9.52 | 13.19 | 6.36
**6.70** | 9.08
**9.63** | 12.92
**14.02** |
| LDPC | (121,60) | 4.82 | 7.21 | 10.87 | 5.22 | 8.31 | 13.07 | 5.17 | 8.31 | 13.30 | 5.74 | 9.26 | 14.78 | 5.63
**6.05** | 8.97
**9.77** | 13.91
**14.92** |
| LDPC | (121,70) | 5.88 | 8.76 | 13.04 | 6.45 | 10.01 | 14.77 | 6.40 | 10.21 | 16.11 | 7.06 | 11.39 | **17.52** | 6.97
**7.42** | 11.17
**12.20** | 14.70
14.92 |
| MacKay | (96,48) | 6.84 | 9.40 | 12.57 | 7.43 | 10.65 | 14.65 | 7.38 | 10.72 | 14.83 | 7.97 | **11.77** | 15.52 | 7.50
**8.43** | 10.97
11.66 | 14.29
**16.08** |
| CCSDS | (128,64) | 6.55 | 9.65 | 13.78 | 7.25 | 10.99 | 16.36 | 6.88 | 10.90 | 15.90 | 7.68 | 11.88 | **17.50** | 7.40
**8.24** | 11.70
**12.36** | 14.76
15.67 |

# 4. Experiments

## 4.1. Training and Testing Setup

The objective of the proposed decoder is to learn direct mapping to the transmitted codewords. For a received signal $\boldsymbol{y}$, we define the binary cross-entropy loss function:

$$\mathcal{L} = -\sum_{i=1}^{n}\{\boldsymbol{x}_i \log(\boldsymbol{p}_i) + (1 - \boldsymbol{x}_i)\log(1 - \boldsymbol{p}_i)\}, \tag{22}$$

where $\boldsymbol{x}_i$ is the $i$-th bit of the transmitted codeword and $p_i$ is our model's predicted probability for that bit. Through backpropagation of this loss, we learn the model's trainable parameters including the input embedding vectors $\{\boldsymbol{w}_{\text{emb},i}\}_{i=1}^{2n-k}$, subspace bases $\boldsymbol{U}_{b,[K]}^l, \boldsymbol{U}_{s,[K]}^l$, dictionaries $\boldsymbol{D}_b^l, \boldsymbol{D}_s^l$, and output projections $\{\boldsymbol{w}_{\text{out},i}\}_{i=1}^{n}$. This direct optimization approach differs from previous methods [15, 17] that predict the multiplicative noise through a preprocessing step.

For all transformer-based models, we set embedding dimension $d = 128$ and number of attention heads $h = 8$. We replace ReLU with GeLU activation [23] in ISTA operations for better gradient flow and smoother optimization landscape. Two configurations are evaluated: WECCT($L = 6$) with 6 layers, and WECCT($L = 12$) with 12 layers to demonstrate the effect of increased optimization steps. The ISTA step size $\eta$ and sparsity weight $\lambda$ are set to 0.1 and 0.5 respectively. We use the Adam optimizer [24] with $\beta_1 = 0.9, \beta_2 = 0.999$ and conduct training for 1000 epochs. Each epoch consists of 1000 minibatches, where each minibatch contains 512 samples. All simulations were conducted using a GPU with 24GB VRAM and over 80 TFLOPS of FP32 compute performance. The training samples $\boldsymbol{y}$ are generated by $\boldsymbol{y} = \boldsymbol{x}_s + \boldsymbol{z}$, where random codewords $\boldsymbol{x}_s$ are transmitted

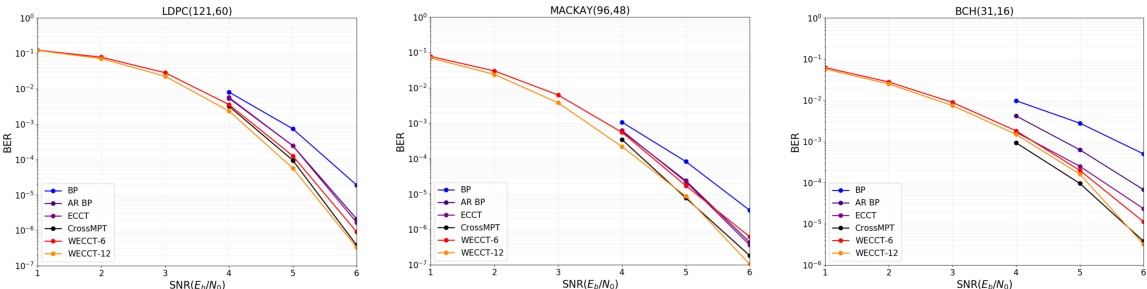

Figure 2: The BER performance of various decoders (BP, AR BP, ECCT, CrossMPT) and WECCT.

through an AWGN channel with noise $z$ sampled from a signal-to-noise ratio (SNR) range of 3 dB to 7 dB. The SNR is measured as energy per bit to noise power spectral density ratio $(E_b/N_0)$, where $E_b$ represents the average energy per information bit and $N_0$ represents the noise power spectral density [1]. The learning rate is initially set to $10^{-4}$ and gradually reduced to $5 \times 10^{-7}$ following a cosine decay scheduler.

To evaluate the effectiveness of WECCT, we conduct extensive experiments on various code families including BCH codes, polar codes and LDPC codes (including MacKay and CCSDS codes). All parity check matrices are taken from [25]. For comparison, we consider the traditional BP decoder with 50 iterations and several neural decoders. Among BP-based neural decoders, we show results for AR BP [26], which has demonstrated superior performance over earlier approaches like Hyper BP [27]. For model-free transformer-based decoders, we compare with ECCT [15] and CrossMPT [17]. During testing, we collect at least 500 frame errors at each SNR value using random codewords, focusing on practical SNR ranges (4, 5, 6 dB). Performance is measured using negative natural logarithm of bit error rate (BER).

## 4.2. Results and Analysis

Our WECCT demonstrates significant parameter efficiency through several theoretically motivated design choices. Compared with ECCT [15] and CrossMPT [17], this dramatic improvement in efficiency comes from our theoretical insights: shared projection matrices in attention reduce parameters from $4d^2$ to $2d^2$ per layer (without sharing between two message-passing iterations), while ISTA network further reduces feedforward parameters from $8d^2$ to $2d^2$ per layer (excluding biases $\mathcal{O}(d)$), complemented by efficient bit/syndrome processing through sparse rate reduction. An analysis of parametric and computational efficiency across different architectures is provided in Appendix C.

Table 1 and Figure 2 show BER performance comparison across different decoders. We see that despite the substantial reduction in parameters, WECCT(L=6) achieves better performance than ECCT across various codes. More importantly, when we increase the number of layers to match the computational budget of previous approaches with WECCT(L=12), our model matches or exceeds CrossMPT's performance while still maintaining a significantly lower parameter count. This scaling behavior provides strong empirical validation for our theoretical framework, where additional layers effectively implement more gradient update steps towards better convergence.

In Appendix D, ablation studies are conducted on the Tanner subspaces mechanism to validate its crucial role in achieving optimal decoding performance. The visualization of rate reduction and sparsification patterns can be found in Appendix E, where we also analyze the rate reduction during training and its correlation with decoding performance. These analyses further support the effectiveness of our framework in achieving the expected theoretical objectives.

## 5. Conclusion

In this paper, we present WECCT, a white-box transformer architecture for error correction code decoding that combines theoretical interpretability with strong empirical performance. By formulating the decoding problem from a sparse coding perspective, we develop a novel MTSA mechanism that explicitly incorporates code structure into representation learning. Our experiments

demonstrate that WECCT achieves competitive performance with significantly fewer parameters than previous approaches, while providing clear theoretical insights into its internal operations. Several promising directions for future work include extending the framework to different channel environments, scaling to longer codes through hierarchical Tanner subspaces, and exploring more targeted sparsification strategies. Through these developments, we believe the white-box approach introduced in this work can lead to efficient neural decoders for modern communication systems while maintaining strong theoretical guarantees.

## Acknowledgements

The research of Shao-Lun Huang is supported in part by the Shenzhen Science and Technology Program under Grant KQTD20170810150821146 and Meituan.

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

## A. Proof of Appproximation 1

*Proof:* For AWGN channels, the ML decoding objective can be written as minimizing the expected squared error between the received signal and the predicted codeword:

$$\hat{\boldsymbol{x}}_s = \operatorname*{argmin}_{\boldsymbol{x} \in \mathcal{C}} \|\boldsymbol{y} - \boldsymbol{x}_{\boldsymbol{s}}\|_2^2. \tag{23}$$

Through Tweedie's formula [20], the optimal estimator can be expressed in terms of the score function:

$$\mathbb{E}[\boldsymbol{x}_s|\boldsymbol{y}] = \boldsymbol{y} + \sigma^2 \frac{\nabla p(\boldsymbol{y})}{p(\boldsymbol{y})} = \boldsymbol{y} + \sigma^2 \nabla \log p(\boldsymbol{y}). \tag{24}$$

In our iterative framework, this denoising process can be formulated as token updates from layer $l$ to layer $l+1$. It moves the bit token set $\boldsymbol{Z}_b^l$ towards the maximum-likelihood token set with respect to the model $\boldsymbol{U}_{[K]}^l$:

$$\boldsymbol{Z}_b^{l+1} = \boldsymbol{Z}_b^l + \sigma^2 \nabla \log p(\boldsymbol{Z}_b^l \mid \boldsymbol{U}_{[K]}^l), \tag{25}$$

where $\boldsymbol{Z}_b^l$ represents bit tokens at layer $l$. One recently popular class of models performing ML estimation is energy-based models [21]. Therefore, following [18], the desired probability distribution of $\boldsymbol{Z}_b$ is known up to constants as

$$p\left(\boldsymbol{Z}_b \mid \boldsymbol{U}_{[K]}\right) = Ce^{-E\left(\boldsymbol{Z}_b \mid \boldsymbol{U}_{[K]}\right)} \doteq C \exp\left(-\lambda\|\boldsymbol{Z}_b\|_1\right) \cdot \frac{\det\left(\boldsymbol{I} + \alpha \boldsymbol{Z}_b^* \boldsymbol{Z}_b\right)}{\prod_{k=1}^K \det\left(\boldsymbol{I} + \beta\left(\boldsymbol{U}_k^* \boldsymbol{Z}_b\right)^*\left(\boldsymbol{U}_k^* \boldsymbol{Z}_b\right)\right)}, \tag{26}$$

where the energy function is defined as

$$E(\boldsymbol{Z}_b \mid \boldsymbol{U}_{[K]}) = -\left[R(\boldsymbol{Z}_b) - R^c(\boldsymbol{Z}_b \mid \boldsymbol{U}_{[K]}) - \lambda\|\boldsymbol{Z}_b\|_1\right]. \tag{27}$$

Note that the term $\det\left(\boldsymbol{I} + \alpha \boldsymbol{Z}_b^* \boldsymbol{Z}_b\right) / \prod_{k=1}^K \det\left(\boldsymbol{I} + \beta\left(\boldsymbol{U}_k^* \boldsymbol{Z}_b\right)^*\left(\boldsymbol{U}_k^* \boldsymbol{Z}_b\right)\right)$ has a natural intrinsic geometric interpretation that it can be regarded as the ratio of the 'volume' of $\boldsymbol{Z}_b$ and the product of 'volumes' of its projections into the subspaces [28].

For ECCs, any valid codeword must satisfy the parity check equations. The syndrome $\boldsymbol{s}(\boldsymbol{y}) = \boldsymbol{H}\boldsymbol{y}_b$ represents a combination of bit values, implying that the induced syndrome tokens $\boldsymbol{Z}_s$ naturally lie in subspaces spanned by bit tokens $\boldsymbol{Z}_b$. Performing joint denoising on syndromes helps better capture parity check constraints and reinforce the structural dependencies among bits in the code, which motivates us to model bits and syndromes together in a unified representation space $\boldsymbol{Z} = [\boldsymbol{Z}_b, \boldsymbol{Z}_s]$. The energy function can then be extended to this joint space:

$$E(\boldsymbol{Z} \mid \boldsymbol{U}_{[K]}) = -\left[R(\boldsymbol{Z}) - R^c(\boldsymbol{Z} \mid \boldsymbol{U}_{[K]}) - \lambda\|\boldsymbol{Z}\|_1\right], \tag{28}$$

Minimizing the energy $E(\boldsymbol{Z} \mid \boldsymbol{U}_{[K]})$ above is equivalent to maximizing the sparse rate reduction objective (4).

## B. Proof of Approximation 2

Here we provide the complete derivation of the MTSA mechanism. The proof follows three main steps: deriving the exact gradient of the coding rate, applying the von Neumann approximation, and incorporating the Tanner graph structure.

*Proof:* Consider the coding rate $R^c(\boldsymbol{Z} \mid \boldsymbol{U}_{[K]})$ where $\boldsymbol{Z} \in \mathbb{R}^{d \times (2n-k)}$ represents the combined bit and syndrome representations. Following the rate reduction framework in [18], the gradient with respect to $\boldsymbol{Z}$ is given by

$$\nabla_{\boldsymbol{Z}} R^c\left(\boldsymbol{Z} \mid \boldsymbol{U}_{[K]}\right) = \beta \sum_{k=1}^K \boldsymbol{U}_k \boldsymbol{U}_k^* \boldsymbol{Z}\left(\boldsymbol{I} + \beta\left(\boldsymbol{U}_k^* \boldsymbol{Z}\right)^*\left(\boldsymbol{U}_k^* \boldsymbol{Z}\right)\right)^{-1}. \tag{29}$$

For computational efficiency, we apply the von Neumann series expansion [29] to approximate the matrix inverse:

$$(\boldsymbol{I} + \beta(\boldsymbol{U}_k^* \boldsymbol{Z})^*(\boldsymbol{U}_k^* \boldsymbol{Z}))^{-1} = \boldsymbol{I} - \beta(\boldsymbol{U}_k^* \boldsymbol{Z})^*(\boldsymbol{U}_k^* \boldsymbol{Z}) + \mathcal{O}(\beta^2). \tag{30}$$

Substituting this approximation into the gradient expression:

$$\begin{aligned}
\nabla_{\boldsymbol{Z}} R^c\left(\boldsymbol{Z} \mid \boldsymbol{U}_{[K]}\right) &\approx \beta \sum_{k=1}^K \boldsymbol{U}_k \boldsymbol{U}_k^* \boldsymbol{Z}\left(\boldsymbol{I} - \beta\left(\boldsymbol{U}_k^* \boldsymbol{Z}\right)^*\left(\boldsymbol{U}_k^* \boldsymbol{Z}\right)\right) \\
&= \beta \sum_{k=1}^K \boldsymbol{U}_k\left(\boldsymbol{U}_k^* \boldsymbol{Z} - \beta \boldsymbol{U}_k^* \boldsymbol{Z}\left[\left(\boldsymbol{U}_k^* \boldsymbol{Z}\right)^*\left(\boldsymbol{U}_k^* \boldsymbol{Z}\right)\right]\right).
\end{aligned} \tag{31}$$

The term $(U_k^* Z)^* (U_k^* Z)$ represents the auto-correlation among projected tokens in the $k$-th subspace. This correlation matrix indicates the subspace memberships and interactions between different tokens. From Definition 1, tokens should only interact within their respective Tanner subspaces defined by $\mathcal{M}(H)$. We incorporate the Tanner graph structure through masked attention:

$$
\begin{aligned}
\nabla_{\mathbf{Z}} R^c \left( \mathbf{Z} \mid \mathbf{U}_{[K]} \right) \approx{} & \beta \sum_{k=1}^{K} \mathbf{U}_k \mathbf{U}_k^* \mathbf{Z} \\
& - \beta^2 \sum_{k=1}^{K} \mathbf{U}_k \left( \mathbf{U}_k^* \mathbf{Z} \operatorname{softmax} \left( (\mathbf{U}_k^* \mathbf{Z})^* (\mathbf{U}_k^* \mathbf{Z}) + \phi(\mathcal{M}(H)) \right) \right),
\end{aligned}
\tag{32}
$$

where $\phi(\mathcal{M}(H))$ is defined in (13). This ensures that token $i$ only attends to token $j$ where $\mathcal{M}(H)_{ji} = 1$. Note that this masking strategy deliberately excludes a token attending to itself in the correlation computation, as our goal is to ensure message passing between bits and syndromes through the Tanner graph structure.

The gradient update can be reformulated into the TSA and MTSA operations. For each subspace basis $\mathbf{U}_k$, the Tanner-subspace Self Attention (TSA) operation is defined as:

$$
\operatorname{TSA}(\mathbf{Z} \mid \mathbf{U}_k) = (\mathbf{U}_k^* \mathbf{Z}) \operatorname{softmax}((\mathbf{U}_k^* \mathbf{Z})^* (\mathbf{U}_k^* \mathbf{Z}) + \phi(\mathcal{M}(H))).
\tag{33}
$$

The Multi-head Tanner-subspace Self Attention (MTSA) operation then aggregates the attention outputs across all subspaces:

$$
\operatorname{MTSA}(\mathbf{Z} \mid \mathbf{U}_{[K]}) = \beta[\mathbf{U}_1, \ldots, \mathbf{U}_K] \begin{bmatrix} \operatorname{TSA}(\mathbf{Z} \mid \mathbf{U}_1) \\ \vdots \\ \operatorname{TSA}(\mathbf{Z} \mid \mathbf{U}_K) \end{bmatrix}.
\tag{34}
$$

This formulation allows us to express the gradient descent update in a concise form:

$$
\mathbf{Z} - \kappa \nabla_{\mathbf{Z}} R^c(\mathbf{Z} \mid \mathbf{U}_{[K]}) \approx (1 - \kappa\beta)\mathbf{Z} + \kappa\beta \operatorname{MTSA}(\mathbf{Z} \mid \mathbf{U}_{[K]}).
\tag{35}
$$

The resulting MTSA mechanism effectively combines the gradient descent on coding rate with the structural constraints of error correction codes. The multi-head design follows similar principles to those in [16], but with a principled interpretation through rate reduction and explicit incorporation of code structure.

## C. Complexity

Table 2: Comparison of parameters and FLOPs for different decoders

| Code | Model | Parameters (M) | FLOPs (M) |
|---|---|---|---|
| LDPC(121,70) | ECCT | 1.23 | 37.7 |
| | CrossMPT | 1.23 | 28.8 |
| | WECCT-6 | 0.46 | 17.2 |
| | WECCT-12 | 0.85 | 33.8 |
| BCH(63,45) | ECCT | 1.19 | 14.0 |
| | CrossMPT | 1.19 | 11.8 |
| | WECCT-6 | 0.43 | 8.4 |
| | WECCT-12 | 0.82 | 16.2 |

The parameter efficiency of WECCT mainly comes from the shared projection design in attention modules. While WECCT uses separate weight matrices for bit-to-syndrome and syndrome-to-bit message passing, it achieves significant parameter reduction by sharing projections between key and value transformations within each attention module, reducing attention parameters from $4d^2$ to $2d^2$ per layer (excluding biases $\mathcal{O}(d)$). Combined with the efficient ISTA operations that replace standard feed-forward networks which reduces parameters from $8d^2$ to $2d^2$ per layer, WECCT achieves

approximately 64% parameter reduction compared to both ECCT and CrossMPT for BCH(63,45) code. Notably, even with doubled layers, WECCT-12 still uses 31% fewer parameters than CrossMPT while achieving comparable or better performance.

In terms of computational complexity, all three models have theoretical complexity $\mathcal{O}(N(d^2(2n - k) + \rho hd))$ with different mask densities $\rho$ [17]. The computation differences mainly come from the attention designs: WECCT further reduces FLOPs by sharing key and value computations in each attention module and using efficient ISTA operations with fewer matrix multiplications than standard feed-forward networks. As shown in Table 2, WECCT-6 requires 40% fewer FLOPs than ECCT and 29% fewer than CrossMPT for BCH(63,45) code. The efficiency gain is even more pronounced for longer codes - for LDPC(121,70), WECCT-6 achieves a 54% FLOPs reduction compared to ECCT and 40% compared to CrossMPT. When doubling the number of decoder layers, the increasing of FLOPs is still acceptable with significantly less parameters.

## D. Ablation Study on Tanner Subspaces Mechanism

Table 3: -ln(BER) results comparing WECCT-6 with and without the Tanner subspaces mechanism for LDPC(121, 60) code

| Model | 4dB | 5dB | 6dB |
|---|---|---|---|
| WECCT-6 | 5.63 | 8.97 | 13.91 |
| WECCT-6 (without Tanner subspaces mechanism) | 3.46 | 4.60 | 6.34 |
| WECCT-12 | 6.05 | 9.77 | 14.92 |
| WECCT-12 (without Tanner subspaces mechanism) | 4.27 | 6.37 | 9.52 |

Tanner subspaces play a crucial role in our framework, ensuring correct subspace compression. Table 3 shows the decoding performance comparing our model with and without the Tanner subspaces mechanism for LDPC(121, 60) code. The performance degradation is substantial when removing the designed mechanism, which clearly validates our architectural choices and provide insights into the model's behavior.

## E. Coding Rate and Sparsity across Layers

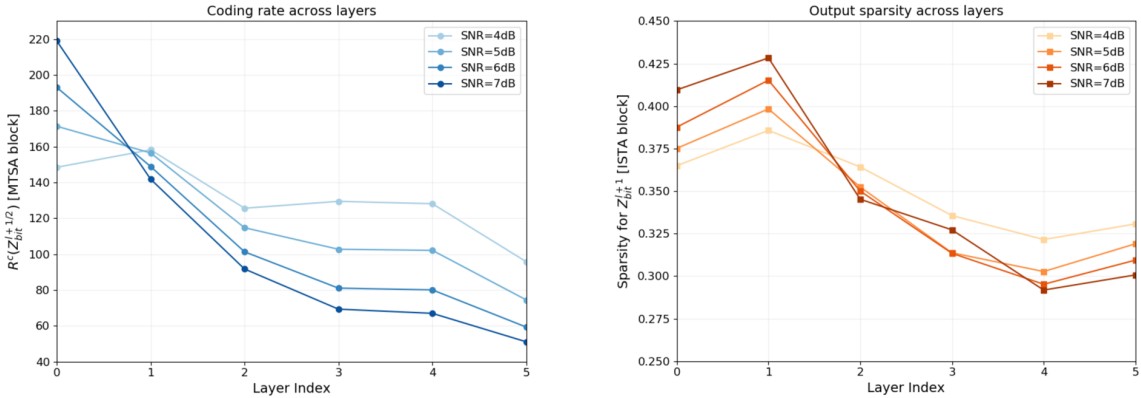

Figure 3: Left: Coding rate reduction across layers for different SNR values. Right: Output sparsification of ISTA blocks across layers.

To validate our theoretical design objectives, we analyze the behavior of both MTSA and ISTA components across network layers. Figure 3 shows that our model successfully achieves the theoretical objectives of rate reduction and structured sparsification.

Specifically, the coding rate $R^c(\boldsymbol{Z}_b^{l+1/2}|\boldsymbol{U}_{b,[K]}^l)$ achieves significant reduction from 219 to 51 at SNR=7dB, demonstrating that our MTSA mechanism effectively compresses the bit representations.

Table 4: Analysis of rate reduction during training WECCT-6 for LDPC(121,60) with SNR=6dB. Values show percentage changes in coding rate relative to the first layer.

| Epochs | -ln(BER) | Layer 2 | Layer 3 | Layer 4 | Layer 5 | Layer 6 |
|--------|----------|---------|---------|---------|---------|---------|
| 1 | 5.27 | +38.19% | +49.57% | +42.62% | +37.58% | +35.11% |
| 100 | 11.41 | -2.17% | -28.20% | -37.71% | -44.69% | -48.03% |
| 200 | 11.58 | -9.51% | -35.31% | -44.40% | -49.09% | -50.12% |
| 350 | 12.44 | -13.48% | -37.20% | -44.20% | -51.95% | -53.39% |
| 1000 | 13.91 | -22.87% | -47.52% | -58.03% | -58.53% | -69.27% |

The sparsification defined as $\|Z_b^{l+1}\|_0/(d \times n)$ across ISTA layers (with ReLU activation) decreases from around 0.425 to 0.3, indicating that ISTA effectively promotes increasingly sparse representations as prescribed by our optimization framework. This trend is particularly pronounced at higher SNR values (6-7dB), suggesting that cleaner channel conditions enable more efficient sparse coding of the representations.

To further validate the connection between rate reduction and decoding performance, we analyzed how coding rate reduction evolves during training. Table 4 shows the relative rate reduction compared to the first layer for LDPC(121,60) with a 6-layer WECCT at SNR=6dB. As training progresses, we simultaneously observe that the decoding performance improves and that the relative rate reduction becomes more significant across layers, suggesting that decoding performance improvements correlate strongly with the increasing rate reduction.

