# OpenReview forum: "White-box Error Correction Code Transformer"
_CPAL.cc/2025/Proceedings_Track — CPAL 2025 (Proceedings Track) Poster_

### Official Review · Reviewer_RvRU · 2025-01-06
**This paper presents the "White-box Error Correction Code Transformer" (WECCT), which enhances the understanding of transformer-based decoding for error correction codes (ECC). By utilizing a novel Multi-head Tanner-subspaces Self Attention mechanism and a dual-stage optimization framework, WECCT integrates ECC structure directly into the learning process. This approach is claimed to be both parametrically efficient and interpretable, achieving competitive performance across various ECC families.**

**Rating:** 7
**Confidence:** 2

**Review:**

The paper innovatively combines white-box modeling with transformer architectures to enhance both the interpretability and efficiency of ECC decoding. And it provides a significant theoretical advancement by linking the decoding process to sparse rate reduction principles, thus grounding the transformer's operations in a solid mathematical framework. Extensive experimental results show that the WECCT outperforms or matches existing state-of-the-art decoders across several ECC families, which substantiates the practical applicability of the proposed model.

While the paper offers comparisons with other methods, it may benefit from wider comparisons across more varied datasets or real-world scenarios to further validate its effectiveness.

---

### Official Review · Reviewer_G27i · 2025-01-11
**Model with good interpretability but less ideal performance than other contemporary method**

**Rating:** 6
**Confidence:** 2

**Review:**

This paper provides a new Transformer structure WECCT to solve the ECC decoding problem. Comparing to previous transformer structure, WECCT adds two unique layers MTSA and ISTA. MTSA, or Multi-head Tanner-subspaces Self Attention block, includes the Tanner graph structure of the code into the calculation of attention, and feed-forward layers promote structured sparsity through Iterative Shrinkage-Thresholding Algorithm (ISTA). The effect of both layers are proved mathematically. This paper provides justifications for the usage of both blocks which ensure the interpretability of the model.

However, the performance of the machine seems sub-optimal comparing to CrossMPT which is proposed in 2024 on majority of the tasks including BCH and Polar.

---

### Official Review · Reviewer_9gYj · 2025-01-19
**Review of  White-box Error Correction Code Transformer**

**Rating:** 6
**Confidence:** 3

**Review:**

While the paper presents interesting theoretical ideas, more work is needed to establish its practical viability and comprehensive understanding:
- While the paper derives its proposal from rate reduction principles mathematically, it does not empirically validate whether rate reduction occurs during training. There is no analysis showing whether better rate reduction correlates with improved decoding performance.
- Lack of ablation studies for key model components such as MTSA and ISTA. To better understand the model's behavior and robustness, it can include an analysis of how different Tanner subspace configurations impact results.
- The evaluation is limited to shorter codes, whereas prior work CrossMPT and ECCT demonstrates effectiveness on much longer codes. This would restricts the practical applicability of WECCT in modern communication standards.

---

### Meta-Review · Area_Chair_fiVU · 2025-02-04

**Recommendation:** Accept (Poster)
**Confidence:** 4

**Metareview:**

The authors present a white box transformer approach for error code correction (EEC). The main idea is to apply a theoretically principled  approach based on sparse rate reduction to come up with an interpretable transformer-based model for EEC. The authors show that the proposed method is parameter efficient and provide several numerical experiments that validate the performance of the proposed approach. During the rebuttal phase, the authors provided detailed responses to reviewers' comments/concers regarding the empirical evaluation of the approach, limitations on length of the codes and comparison to other SOTA approaches. Despite some limitations that should be addressed in future extensions of this work (such as adaptting the approach to medium/long codes, further improving empirical performance compared to SOTA), the contributions of the current paper are important and therefore I recommend accepting it to CPAL.

---

### Decision · Program_Chairs · 2025-02-11

Accept (Poster)